## PERSPECTIVE

### An ADP getaway service, not an ATP tank: Insights into the phosphagen system in neurons

Madeline P. Marques[1]
and Kartik Venkatachalam[1,2,3]

[1]*Department of Integrative Biology and Pharmacology, McGovern Medical School at the University of Texas Health Sciences Center (UTHealth), Houston, TX, USA*

[2]*Neuroscience Graduate Program, The University of Texas MD Anderson Cancer Center UTHealth Houston Graduate School of Biomedical Sciences, Houston, TX, USA*

[3]*Molecular and Translational Biology Graduate Program, The University of Texas MD Anderson Cancer Center UTHealth Houston Graduate School of Biomedical Sciences, Houston, TX, USA*

Email: Madeline.P.Marques@uth.tmc.edu and kartik.venkatachalam@uth.tmc.edu

Handling Editors: Katalin Toth & Matthew Fogarty

The peer review history is available in the Supporting Information section of this article (https://doi.org/10.1113/JP290372#support-information-section).

Historical understanding of how cells meet sudden surges in energy demand was shaped by studies in muscle, a choice that made intuitive sense given that muscle cells rapidly produce large amounts of ATP to support bouts of intense physical activity. Foundational studies led to the discovery of the phosphagen system, comprising creatine kinase in vertebrates and ArgK in many invertebrates (Wallimann et al., 2011). These kinases rapidly convert ADP into ATP using high-energy phosphate donors, thereby buffering ATP levels when glycolysis and mitochondrial oxidative phosphorylation (OXPHOS) cannot keep pace with the energy demand (Wallimann et al., 2011). Classical descriptions frame phosphagens as 'temporal energy buffers' as they provide ATP exactly *when* needed (Newsholme et al., 1978). It is now evident that the physical proximity of phosphagen kinases to sites of energy consumption also permits their roles as 'spatial energy buffers' that overcome limitations in ATP and ADP diffusion, ensuring that ATP appears exactly *where* it is needed (Meyer et al., 1986).

Considering the steep metabolic requirements of neurons, surprisingly little is known about how these cells use the phosphagen system to accommodate their highly volatile energy demands. Writing in this issue of *The Journal of Physiology*, Justs *et al.* demonstrate that motor neurons also rely on the phosphagen system to support epochs of intense energy demand (Justs et al., 2026), but not in the way one might assume. Employing state-of-the-art metabolite biosensor imaging, electrophysiology and computational modelling, the authors demonstrate that the phosphagen system in *Drosophila* motor neurons is required specifically to maintain neurotransmitter release during high-frequency stimulation (Justs et al., 2026). The authors knockdown *ArgK1*, the gene encoding the primary phosphagen kinase in *Drosophila*. Terminals in *ArgK1* knockdown neurons compensated by augmenting glycolysis, yet could not sustain neurotransmission at high firing frequencies (Justs et al., 2026). Strikingly, *ArgK1* knockdown affected neither presynaptic $Ca^{2+}$ extrusion during burst firing, nor the endurance to induced muscle contractions (Justs et al., 2026). These data reveal a division of labour, with the phosphagen system sustaining vesicle release during energy-intensive firing, and OXPHOS being sufficient for $Ca^{2+}$ handling (Chouhan et al., 2012; Justs et al., 2026).

To probe the mechanism, the authors incorporated a biophysical model that simulates ATP dynamics under different firing parameters. Consistent with the absence of effects on $Ca^{2+}$ extrusion dynamics, removal of the phosphagen system from the model had minimal effect on ATP levels, even during 60 Hz stimulation (Justs et al., 2026). Instead, loss of the phosphagen system, caused a pronounced drop in the ATP/ADP ratio (Justs et al., 2026). Thermodynamically, this matters because the free energy of ATP hydrolysis ($\Delta G$) depends on the ATP/ADP ratio, and not on ATP abundance. When ADP accumulates, $\Delta G$ becomes less negative, pushing the ATP–ADP reaction closer to equilibrium and reducing the energy available to perform biological work.

In other words, ATP levels look 'normal', although the cell becomes energetically less capable.

The key insight emerging from this work is that the phosphagen system does not simply regenerate ATP, but continuously *extracts* ADP from the sites of ATP hydrolysis (Justs et al., 2026). By keeping local ADP levels low, ArgK maintains a higher ATP/ADP ratio, thus holding $\Delta G$ farther from equilibrium and preserving the thermodynamic driving force for work. Rather than an ATP reservoir, the phosphagen system functions more as an ADP clearance mechanism. Because the diffusion of ADP in the cytosol is effectively lower than that of ATP, the accumulation of ADP represents a major bioenergetic challenge during periods of ATP hydrolysis. The demonstration that the phosphagen system contributes to ADP clearance and prevents the thermodynamic compression associated with the drop in ATP/ADP ratio is a major step forward in our understanding of how neurons respond to periods of high energy demand.

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

## Additional information

### Competing interests

The authors declare no competing interests.

### Author contributions

M.M. and K.V.: conception or design of the work; drafting the work or revising it critically for important intellectual content. Both authors have read and approved the final version of this manuscript and agree to be accountable for all aspects of the work in ensuring that questions related to the accuracy or integrity of any part of the work are appropriately investigated and resolved. All persons designated as authors qualify for authorship, and all those who qualify for authorship are listed.

### Funding

The work in the Venkatachalam lab was supported by National Institutes of Health (NIH) grants R01AG069076, R01AG072176, and R21AG087381 to K.V.

### Keywords

ATP/ADP ratio, bioenergetics, *Drosophila*, motor neuron, phosphagen

## Supporting information

Additional supporting information can be found online in the Supporting Information section at the end of the HTML view of the article. Supporting information files available:

**Peer Review History**

