## [Peer Review History · The Journal of Physiology]

An ADP getaway service, not an ATP tank—insights into the phosphagen system in neurons

Madeline P Marques and Kartik Venkatachalam
DOI: 10.1113/JP290372

Corresponding author(s): Kartik Venkatachalam (kartik.venkatachalam@uth.tmc.edu)

Review Timeline:

Submission Date:	07-Nov-2025
Editorial Decision:	18-Nov-2025
Revision Received:	24-Nov-2025
Editorial Decision:	05-Dec-2025
Revision Received:	08-Dec-2025
Accepted:	23-Jan-2026

Senior Editor: Katalin Toth

Reviewing Editor: Matthew Fogarty

Transaction Report:

Dear Dr Venkatachalam,

Re: JP-P-2025-290372 "**An ADP getaway service, not an ATP tank-insights into the phosphagen system in neurons**"
by Madeline P Marques and Kartik Venkatachalam

Thank you for submitting your manuscript to The Journal of Physiology. It has been assessed by a Reviewing Editor and by 1 expert referee and we are pleased to tell you that it is acceptable for publication following satisfactory revision.

The review comments are copied at the end of this email.

Please address all the points raised and incorporate all requested revisions or explain in your Response to Referees why a change has not been made. We hope you will find the comments helpful and that you will be able to return your revised manuscript within 2 weeks. If you require longer than this, please contact journal staff: jp@physoc.org.

LANGUAGE EDITING AND SUPPORT FOR PUBLICATION: If you would like help with English language editing, or other article preparation support, Wiley Editing Services offers expert help, including English Language Editing, as well as translation, manuscript formatting, and figure formatting at www.wileyauthors.com/eoo/preparation. You can also find resources for Preparing Your Article for general guidance about writing and preparing your manuscript at www.wileyauthors.com/eoo/prepresources.

REVISION CHECKLIST:

We look forward to receiving your revised submission.

Yours sincerely,

Katalin Toth
Senior Editor
The Journal of Physiology

REQUIRED ITEMS

1) - Please include a full, separate title page as part of your main article (Word) file, which should contain the following: title, authors, affiliations, corresponding author name and contact details, keywords, and running title.

EDITOR COMMENTS

Reviewing Editor:

The Minor corrections and citation should be all this Viewpoint needs.

REFEREE COMMENTS

Referee #1:

The only suggestion we would make is to reword "Because the diffusion of ADP in the cytosol is inherently lower than that of ATP" to "Because the diffusion of ADP in the cytosol is effectively lower than that of ATP". We are suggesting this change to avoid giving the impression that the diffusion coefficient is smaller for ADP, as the modeling of Yoshikazi et al (1990) gives ADP and ATP the same diffusion coefficient. What Yoshikazi et al (1990) do say is that ADP doesn't move very far compared to ATP in the same amount of time, because ADP gets quickly rounded up more quickly. Therefore, without getting into this detail, the word "effectively" is better able to accommodate the nuance.

END OF COMMENTS

Department of Integrative Biology and Pharmacology

Kartik Venkatachalam, Ph.D.

Professor

Director, Neuroscience Program,

The University of Texas MD Anderson Cancer Center UTHealth Houston Graduate School of Biomedical Sciences

Room 4.214 MSB, 6431 Fannin Street, Houston TX 77030, USA

Phone: (713) 500-7504, Fax: (713) 500-7456

<https://med.uth.edu/ibp/faculty/kartik-venkatachalam/>

<http://www.utflylab.com>

November 24, 2025

Dear editors,

We would like to submit our revised manuscript titled, “**An ADP getaway service, not an ATP tank—insights into the phosphagen system in neurons**” for publication as a perspective in *J. Physiology*. This manuscript is focused on JP-RP-2025-288916R1 "Optimal Neuromuscular Performance Requires Motor Neuron Phosphagen Kinases" by Karlis A Justs, Danielle V Latner, Carlos D Oliva, Yosuf Arab, Gabriel G Bonassi, Olena Mahneva, Sarah Crill, Sergio Sempertegui, Paul A Kirchman, Yaouen Fily, and Gregory T Macleod. We have made the requested minor textual correction and have updated the title page to include all the information requested in your email. Please note that we have have not referenced the focus paper as we do not have the pertinent citation information. I anticipate we can add this reference prior to publication.

Sincerely,

Kartik Venkatachalam, Ph.D.
Professor of Integrative Biology and Pharmacology

Dear Dr Venkatachalam,

Re: JP-P-2025-290372R1 "**An ADP getaway service, not an ATP tank-insights into the phosphagen system in neurons**" by Madeline P Marques and Kartik Venkatachalam

Thank you for submitting your manuscript to The Journal of Physiology. It has been assessed by a Reviewing Editor and we are pleased to tell you that it is acceptable for publication following satisfactory revision.

You can update the reference to Justs on the article proofs.

The review comments are copied at the end of this email.

Please address all the points raised and incorporate all requested revisions or explain in your Response to Referees why a change has not been made. We hope you will find the comments helpful and that you will be able to return your revised manuscript within 2 weeks. If you require longer than this, please contact journal staff: jp@physoc.org.

REVISION CHECKLIST:

We look forward to receiving your revised submission.

Yours sincerely,

Katalin Toth
Senior Editor
The Journal of Physiology

REQUIRED ITEMS

EDITOR COMMENTS

Reviewing Editor:

I thank the authors for adding the title page and keyword info.

The citation for the focus paper does not need full pagination volume - please add using your reference manager software.

Title: Optimal Neuromuscular Performance Requires Motor Neuron Phosphagen Kinases

Authors: Karlis A Justs, Danielle V Latner, Carlos D Oliva, Yosuf Arab, Gabriel G Bonassi, Olena Mahneva, Sarah Grill, Sergio Sempertegui, Paul A Kirchman, Yaouen Fily, and Gregory T Macleod

Journal: J Physiol

Year: 2025

REFEREE COMMENTS

END OF COMMENTS

Department of Integrative Biology and Pharmacology

Kartik Venkatachalam, Ph.D.

Professor

Director, Neuroscience Program,

The University of Texas MD Anderson Cancer Center UTHealth Houston Graduate School of Biomedical Sciences

Room 4.214 MSB, 6431 Fannin Street, Houston TX 77030, USA

Phone: (713) 500-7504, Fax: (713) 500-7456

<https://med.uth.edu/ibp/faculty/kartik-venkatachalam/>

<http://www.utflylab.com>

December 8, 2025

Dear editors,

We would like to submit our revised manuscript titled, “**An ADP getaway service, not an ATP tank—insights into the phosphagen system in neurons**” for publication as a perspective in *J. Physiology*. This manuscript is focused on JP-RP-2025-288916R1 "Optimal Neuromuscular Performance Requires Motor Neuron Phosphagen Kinases" by Karlis A Justs, Danielle V Latner, Carlos D Oliva, Yosuf Arab, Gabriel G Bonassi, Olena Mahneva, Sarah Crill, Sergio Sempertegui, Paul A Kirchman, Yaouen Fily, and Gregory T Macleod. We have now referenced the original focus paper, as requested.

Sincerely,

Kartik Venkatachalam, Ph.D.
Professor of Integrative Biology and Pharmacology

Dear Dr Venkatachalam,

Re: JP-P-2025-290372R2 "**An ADP getaway service, not an ATP tank-insights into the phosphagen system in neurons**" by Madeline P Marques and Kartik Venkatachalam

We are pleased to tell you that your paper has been accepted for publication in The Journal of Physiology.

Please note that Perspective articles are not typically covered by institutional open access agreements with our publisher, Wiley. Wiley do not offer article processing charge (APC) discounts for smaller article types in hybrid subscription journals, meaning that if you wish for your Perspective to be published Open Access, you will have to pay the full APC. As such, we recommend authors publish Perspectives 'behind the paywall', where they will become freely accessible after a 12-month embargo (i.e. please select the NON open access option via Wiley Author services during proofing).

Should you wish to pay for Open Access, you will be able to place an order by logging into Wiley Author services.

Yours sincerely,

Katalin Toth
Senior Editor
The Journal of Physiology

IMPORTANT POINTS TO NOTE FOLLOWING ACCEPTANCE OF YOUR PAPER:

- **IMPORTANT NOTICE ABOUT OPEN ACCESS:** To assist authors whose funding agencies mandate immediate public access to published research findings, The Journal of Physiology allows authors to pay an Open Access (OA) fee to have their papers made freely available immediately on publication.

- You can help your research get the attention it deserves! Check out Wiley's free Promotion Guide for best-practice recommendations for promoting your work at: www.wileyauthors.com/eoo/guide. You can learn more about Wiley Editing Services which offers professional video, design, and writing services to create shareable video abstracts, infographics, conference posters, lay summaries, and research news stories for your research at: www.wileyauthors.com/eoo/promotion.

- If you would like to receive our 'Research Roundup', a monthly newsletter highlighting the cutting-edge research published in The Physiological Society's family of journals (The Journal of Physiology, Experimental Physiology, Physiological Reports, The Journal of Nutritional Physiology and The Journal of Precision Medicine: Health and Disease), please click this link, fill in your name and email address and select 'Research Roundup':